# Metronomic Chemotherapy in Prostate Cancer

**DOI:** 10.3390/jcm11102853

**Published:** 2022-05-18

**Authors:** Piotr J. Wysocki, Maciej T. Lubas, Malgorzata L. Wysocka

**Affiliations:** 1Department of Oncology, Medical College, Jagiellonian University, 30-252 Krakow, Poland; 2Department of Oncology, Krakow University Hospital, 30-688 Krakow, Poland; mlubas@su.krakow.pl; 3Oncoaid, ul. Sliska 7/2, 30-504 Krakow, Poland

**Keywords:** metronomic chemotherapy, prostate cancer, CRPC, castration, cyclophosphamide, capecitabine, paclitaxel, immunomodulation, angiogenesis, microbiome

## Abstract

Despite the significant expansion of the therapeutic armamentarium associated with the introduction of novel endocrine therapies, cytotoxic agents, radiopharmaceuticals, and PARP inhibitors, progression of metastatic castration-resistant prostate cancer (mCRPC) beyond treatment options remains the leading cause of death in advanced prostate cancer patients. Metronomic chemotherapy (MC) is an old concept of wise utilization of cytotoxic agents administered continuously and at low doses. The metronomic is unique due to its multidimensional mechanisms of action involving: (i) inhibition of cancer cell proliferation, (ii) inhibition of angiogenesis, (iii) mitigation of tumor-related immunosuppression, (iv) impairment of cancer stem cell functions, and (v) modulation of tumor and host microbiome. MC has been extensively studied in advanced prostate cancer before the advent of novel therapies, and its actual activity in contemporary, heavily pretreated mCRPC patients is unknown. We have conducted a prospective analysis of consecutive cases of mCRPC patients who failed all available standard therapies to find the optimal MC regimen for phase II studies. The metronomic combination of weekly paclitaxel 60 mg/m^2^ i.v. with capecitabine 1500 mg/d p.o. and cyclophosphamide 50 mg/d p.o. was selected as the preferred regimen for a planned phase II study in heavily pretreated mCRPC patients.

## 1. Standard Systemic Treatment of Advanced Prostate Cancer

Endocrine therapy, basically androgen deprivation (ADT), is the standard treatment of prostate cancer (PC). This neoplasm arises from prostate gland epithelial cells located in lobules and ducts, whose growth and differentiation are tightly regulated by androgens, mainly testosterone. Therefore, well-differentiated prostate cancers are usually susceptible to androgen deprivation which allows for long-term inhibition of disease progression. However, many prostate cancers are poorly differentiated (Gleason ≥ 8 or grading ≥ 4), and therefore the sensitivity to endocrine treatment is low and short. Historically, surgical castration, and later, pharmacological castration, represented the only option of systemic treatment with proven activity in prostate cancer, and chemotherapy was generally assumed as not active in this disease [1]. The development of resistance to androgen deprivation (castration) is usually associated with increased expression and/or hypersensitivity of androgen receptors in PC cells and autocrine and paracrine production of androgens beyond the control hypothalamus–pituitary–gonadal axis [2,3]. Therefore, since, at the initial stage of castration resistance, PC cells demonstrate increased instead of decreased endocrine sensitivity, novel generations of hormonal agents like abiraterone acetate or irreversible androgen receptor (AR) blockers (enzalutamide, daralutamide, and apalutamide) could show high anticancer activity leading to prolonged disease control and improved patients’ outcomes [4,5,6,7,8]. However, prostate cancer cells will ultimately become resistant to endocrine treatment by activating various hormone-unrelated molecular mechanisms, leading to increased aggressiveness and proliferation. However, these features of castration-resistant prostate cancer reflect increased sensitivity of PC cells to antiproliferative cytotoxic agents such as microtubule inhibitors. Docetaxel was the first chemotherapy drug shown, in 2004, to improve the overall survival of metastatic castration-resistant prostate cancer (mCRPC) patients [9]. Additionally, due to its inhibitory impact on AR transcriptional activity, docetaxel’s antitumor potential was also confirmed in metastatic castration-sensitive PC (mCSPC), where it was combined in first-line of treatment with androgen deprivation [10,11]. The increased chemotherapy sensitivity of CRPC cells, which results from AR-independency and increased proliferation rate, is not restricted to a single class of cytotoxic drugs (microtubule depolymerization inhibitors—taxanes). In hormone-refractory PC, various cytotoxic agents have demonstrated clinical activity, but only large phase III clinical studies on taxanes had sufficient power to prove a significant improvement of outcomes with the use of chemotherapy.

With the widespread of novel, highly active endocrine agents, the incidence of a previously rare phenomenon of neuroendocrine differentiation increased significantly [12]. Cells with neuroendocrine features occur in approx. 1% of primary prostate tumors, but neuroendocrine differentiation is present in 25% of end-stage metastatic castration-resistant disease cases [13]. The loss of adenocarcinoma phenotype and transformation into small cells with neuroendocrine features is one of the ways the androgen-deprived PC utilizes to get rid of androgen dependency. Characteristic features of small-cell neuroendocrine tumors are high aggressiveness, rapid proliferation, and sensitivity to chemotherapy platinum-based regimens usually used to treat neuroendocrine cancers such as small-cell lung cancer [14]. However, the sensitivity of prostate cancer to platinum compounds, which is atypical for this tumor besides neuroendocrine differentiation, can also result from prostate cancer genetic background. Inherited mutations of genes responsible for homologous recombination (homologous recombination deficiency—HRD) are relatively rare, with germinal mutations of the two most important cancer predisposition genes—*BRCA1* or *BRCA2* found in 0.4% and 1–2% of PC patients, respectively [15,16]. On the other hand, somatic or germinal mutations in genes responsible for double-strand DNA repair can be detected in 11.8% of tumor tissues from mCRPC patients [17]. Moreover, in a screening phase of a phase III study evaluating olaparib in mCRPC patients, at least one mutation/deletion (germline or somatic) within 15 genes responsible for homologous recombination was detected in 28% of patients (778 of 2792 tumors with HRD) [18]. Deleterious mutations in *BRCA1* and *BRCA2* genes in prostate cancer cells are predictive of the olaparib activity, which has been shown to significantly improve the overall survival of advanced mCRPC patients. However, recent data suggest the presence of HRD is also predictive of the activity of DNA-damaging cytotoxic agents. Therefore, in heavily pretreated mCRPC patients alkylating agents such as platinum compounds or cyclophosphamide can be more effective than previously, based on data from clinical studies conducted 2–3 decades ago, considered.

## 2. Challenges Associated with the Use of Standard Chemotherapy in Prostate Cancer

Prostate cancer is relatively resistant to chemotherapy. Before the introduction of docetaxel, the only available cytotoxic drug with any activity against prostate cancer was mitoxantrone [19]. However, when combined with prednisone, the benefit of this topoisomerase II inhibitor was superior to prednisone alone only in terms of pain control and quality of life, but not in terms of overall survival. In 2004, docetaxel was approved for the treatment of mCRPC patients based on two landmark studies—TAX327 and SWOG 9916, which demonstrated significant improvement in OS with docetaxel-based regimens compared to mitoxantrone [9,20]. Particularly, in TAX327 study, which compared 3-weekly docetaxel (75 mg/m^2^ q3w) with weekly docetaxel (30 mg/m^2^ q1w) and with mitoxantrone (all drugs combined with prednisone 10 mg/daily), the 3-weekly regimen significantly improved median OS compared to mitoxantrone (HR = HR = 0.79, 95%CI 0.67–0.93) [9]. The median OS was 19.2, 17.8, and 16.3 months, in docetaxel q3w, docetaxel q2w, and mitoxantrone arms, respectively. Additionally, both docetaxel-based regimens significantly improved patients’ quality of life (QOL) compared to mitoxantrone [21]. However, the significant improvement of outcomes with docetaxel administered at 3-week intervals came at the cost of increased toxicity. Grade ≥ 3 adverse events such as neutropenia, fatigue, diarrhea, neuropathy, stomatitis, dyspnea, and peripheral edema were significantly more frequent in docetaxel q3w than mitoxantrone arm.

The toxicity of standard docetaxel dose can be reduced by administering the drug in shorter intervals at a reduced dose. The TAX327 study has not confirmed the clinical utility of weekly docetaxel at 30 mg/m^2^, but a phase III study of a Scandinavian PROSTY group demonstrated at least similar activity of biweekly docetaxel 50 mg/m^2^ compared to the standard dosing. The biweekly administration of docetaxel was associated with significant improvement of the primary endpoint, which was time-to-treatment failure (TTF), compared to the 3-weekly regimen [22]. The median TTF was 5.6 months vs 4.9 months, for 2-weekly and 3-weekly regimens, respectively. Typical adverse events occurred less frequently with the lower, biweekly dose of docetaxel. Unexpectedly, secondary endpoints such as OS and PFS were significantly improved with biweekly docetaxel. There were statistically significant differences in overall QOL values and pain values favoring the biweekly treatment arm regarding patient-reported outcomes [23].

Over time, despite numerous attempts to improve the activity of docetaxel-based chemotherapy regimens by combining them with targeted agents, 3- or 2-weekly monotherapy with docetaxel remained the chemotherapy of choice for mCRPC patients [24,25,26,27,28].

Another taxane, cabazitaxel, is the second and the last cytotoxic agent with proven, significant activity in patients with advanced, castration-resistant prostate adenocarcinoma. A phase III TROPIC study comparing cabazitaxel 25 mg/m^2^ q3w (+prednisone) with mitoxantrone (+prednisone) in mCRPC patients following failure of docetaxel-based chemotherapy demonstrated significant superiority of cabazitaxel [29]. The median OS and PFS were 15.1 months vs. 12.7 months and 2.8 months vs 1.4 months in cabazitaxel and mitoxantrone arms, respectively. The hazard ratio for death of men treated with cabazitaxel compared with those taking mitoxantrone was 0.70 (95% CI 0.59–0.83). However, again the improvement in outcomes came at the cost of increased toxicity; 5% of patients treated with cabazitaxel and 2% treated with mitoxantrone died within 30 days of the last infusion. Still, none of the deaths in the cabazitaxel arm was associated with disease progression. The most frequent adverse events were associated with hematological toxicity (neutropenia, leukopenia, and anemia), and neutropenic fever occurred in 8% of patients. Subsequently, a non-inferiority phase III PROSELICA study aimed to compare a lower dose of cabazitaxel (20 mg/m^2^ q3w) with the standard dose [30]. The non-inferiority endpoint of at least 50% of OS benefit with low-dose cabazitaxel compared to the standard dose was achieved. Grade ≥ 3 adverse events were 39.7% for the low-dose cabazitaxel and 54.5% for the standard dose. Health-related quality of life did not differ between cohorts. However, significant differences were observed in favor of standard-dose cabazitaxel for PSA response and time to PSA progression (HR = 1.195; 95% CI 1.025 to 1.393).

There is no doubt that improvement in outcomes of prostate cancer patients achieved with taxane-based chemotherapy regimens came at the cost of toxicity, which, especially in elderly and fragile patients, may represent a significant drawback. However, attempts to optimize chemotherapy with dose reduction or decreased intensity may lead to inferior outcomes in at least a fraction of mCRPC patients in a real-world setting. Moreover, most mCRPC patients treated in clinical practice are old and present with a deteriorating performance status due to advanced disease and long-term systemic treatment. Therefore, after docetaxel treatment failure, many of them are no more candidates for further, intensive anticancer treatment, including but not restricted to cabazitaxel-based therapy. Since there is no further treatment option for many of such patients but hospice care, novel, low-toxic therapeutic approaches that would give a chance to improve survival and maintain quality of life, such as metronomic chemotherapy, are urgently needed.

## 3. Metronomic Chemotherapy—Mechanisms of Action

Metronomic chemotherapy (MC) is a concept of continuous administration of cytotoxic drugs at low doses. Unlike standard chemotherapy regimens, which usually use maximal-tolerated doses (MTD) of chemotherapeutics and require long (usually >2 weeks) recovery periods, the metronomic chemotherapy safety profile allows for continuous, very frequent drug administration [31,32]. Metronomic chemotherapy is mainly based on oral drugs, which patients can take even several times a day. Standard chemotherapy regimens based on MTD are very useful for treating aggressive cancers with a high proliferation rate [33,34]. Highly proliferating tumors, in which 90–100% of cells are actively dividing, can wholly and rapidly respond to aggressive chemotherapy. However, the usual proliferation rate of most solid tumors in adults is lower, and Ki67 scores are far below 50% (often below 15% [35,36]), which means that far beyond 50% of tumor cells (non-proliferating cells) can survive administration of chemotherapy at MTD at wide intervals. In order to control such slowly proliferating tumors, tumor cells must be exposed continuously to antiproliferative agents which will impede cellular division whenever it occurs. It is thus clear that such a long-term exposure of tumor cells to anticancer agents can only be achieved by continuous administration of cytotoxic drugs. However, the MC not only means continuous inhibition of tumor-cell proliferation but also activation of other, clinically essential mechanisms of action, which represent crucial hallmarks of metronomic chemotherapy.

### 3.1. Inhibition of Angiogenesis

Angiogenesis is a crucial step in tumor development since its overgrowth beyond 2–3 mm in diameter must be associated with the development of tumor-associated vasculature [37]. The angiogenic switch is related to releasing proangiogenic factors by hypoxic cancer cells, which stimulate endothelial cells in nearby blood vessels to proliferate and migrate towards the tumor center. Some initial observations suggested that many cytotoxic drugs could inhibit the activation of endothelial cells, but the effect subsided during the withdrawal of medications in treatment-free intervals [38]. Therefore, the antiangiogenic potential of standard MTD-based chemotherapies is, at best, weak and intermittent. In contrast, metronomic chemotherapy administered continuously exerts continuous antiproliferative activity on endothelial cells, thus inhibiting their potential to create pathological vasculature. Numerous studies showed that metronomic chemotherapy inhibited the proliferation and circulation of endothelial cells and endothelial progenitor cells (CEPs) and reduced the differentiation of immature endothelial cells [39,40,41]. Moreover, MC can inhibit the secretion of proangiogenic factors by directly inhibiting hypoxia-inducible factor 1α (HIF1α) activity in tumor cells while simultaneously increasing the production of antiangiogenic molecules. It has been recently demonstrated that MTD-based chemotherapy can promote metastatic dissemination via activation of a transcriptional program dependent on HIF-1α and HIF-2α, and that this effect could be mitigated by switching from MTD to metronomic chemotherapy [42]. Bocci et al. demonstrated that metronomically administered cyclophosphamide inhibits the synthesis of thrombospondin-1 (TSP-1) and directly mediates growth arrest and apoptosis of endothelial cells.

Additionally, they showed the inhibition of angiogenesis in human tumor-bearing immunodeficient mice treated with low daily doses of CTX [43]. Later, it was demonstrated by several groups that the antiangiogenic effect is not restricted to cyclophosphamide but can be observed with many chemotherapy drugs administered in a metronomic manner [44,45]. Over the years, it turned out that metronomic chemotherapy stimulates the production of numerous antiangiogenic factors such as: endostatin, angiostatin, soluble VEGF receptors (sVEGFRs), and pigment epithelium-derived factor (PEDF) [46].

### 3.2. Immunomodulation

The immunological effects of chemotherapeutic drugs are extremely diverse and complex [47,48]. The classical assumption was that standard MTD-based chemotherapy is highly immunosuppressive, owning to its myelosuppressive activity. However, many studies suggested that chemotherapy-induced myelosuppression correlates with improved outcomes [49]. In a retrospective analysis of the TROPIC trial, severe neutropenia in mCRPC patients with initially high NLR (neutrophil-to-lymphocyte ratio) treated with cabazitaxel was associated with significantly improved OS and PFS [50]. It is, however, unknown whether these improved outcomes resulted from depletion of immunosuppressive mononuclear cells or whether the neutropenia only reflected the high cytotoxic activity of cabazitaxel in particular patients. On the other hand, a significant amount of data supports the beneficial impact of metronomic chemotherapy on anticancer immune responses. The immunostimulatory effects of metronomic chemotherapy include (i) induction of immunogenic cancer cell death [51], (ii) preferential depletion of regulatory T (Treg) cells [52,53,54,55], (iii) enhancement of antigen-presentation through stimulation of dendritic cells activity (DC) [56] and increased immunogenicity of cancer cells [57], (iv) inhibition of myeloid-derived suppressor cells (MDSC) [58], and (v) activation of immune effector cells, such as tumor-specific T cells [59,60] and γδT cells [61].

### 3.3. Targeting Cancer Stem Cells

Cancer stem cells (CSC) represent a population of cancer-initiating cells characterized by a slow proliferation rate, self-renewal capability, and resistance to chemotherapy and irradiation. In various in vitro models, metronomic chemotherapy impeded the survival and proliferation of CSCs. In a pancreatic tumor xenograft model, metronomic administration of cyclophosphamide reduced the number of CD133+ precursor cells and CD133+/CD44+/CD24+ cancer stem cells [62]. Similarly, metronomic, but not MTD-based, cyclophosphamide inhibited the function of C6 rat glioma CSCs [63]. In a preclinical model of luminal breast cancer, metronomic paclitaxel reduced the population of CD44+/CD24- stem cells [64]. Expansion of stem-like tumor-initiating cells correlates with the development of chemoresistance and metastatic potential of cancer cells. In animal models, MTD chemotherapy, regardless of cytotoxic agents used, induced persistent STAT-1 and NF-κB activity in carcinoma-associated fibroblasts leading to secretion of CXCR2-activating chemokines [65]. Stimulation of CXCR2 on cancer cells converts them into tumor-initiating CSC responsible for invasive behaviors and paradoxically enhanced tumor aggression after therapy. In contrast, the same overall dose of cytotoxic agents administered as an MC regimen prevented therapy-induced CXCR2 paracrine signaling, thus enhancing treatment response and extending the survival of tumor-bearing mice [65].

### 3.4. Modulation of Gut and Tumor Microbiome

Over the last few years, increasing attention has been paid to the host gut microbiome defined as the collection of genomes from all microorganisms in the gastrointestinal tract, especially within the intestines. However, it has been demonstrated that the microbiome in general (not only the gut, but also local or even intratumoral microbiome) is associated with the pathogenesis of various cancers such as colorectal, breast, ovarian, lung, and prostate cancers [66,67,68,69,70]. For example, compared to a healthy population, the fecal gut microbiome in CRC patients is enriched with *Prevotella*, *Collinsella*, and *Peptostreptococcus* and contained significantly lower concentrations of Escherichia-Shigella [70]. The gut microbiota of patients diagnosed with PC on transrectal biopsy was significantly enriched with *Bacteroides* spp. compared to patients with negative biopsy results [71]. In prostate cancer, the microbiota of tumor and peri-tumoral regions had higher relative abundance of *Staphylococcus* spp. than normal areas. Still, the normal areas had a higher abundance of *Streptococcus* spp. than the tumor and the peri-tumoral regions [72].

With the advent of novel immunotherapies, the role of the microbiome has become even more important in the context of its significant role in shaping responses to checkpoint inhibitors. The gut microbiota plays an important role in regulating systemic immune responses [73,74,75], and the role of intestinal microbiota in mediating immune activation in response to chemotherapeutic agents has been clearly demonstrated [76,77]. In an animal melanoma model, the presence of *Bifidobacterium* correlated with the activation of antitumor immune mechanisms by anti-PD1 checkpoint inhibitors, and oral administration of *Bifidobacterium* alone improved tumor control to the same degree as PD-L blockade [78]. Moreover, a combination of anti-PD-L1 antibody with oral administration of *Bifidobacterium* completely blocked the growth of melanoma tumors in mice. Many studies showed that administration of wide-spectrum antibiotics before initiation of treatment with checkpoint inhibitors could completely abolish the efficacy of immunotherapy in patients with various tumors [79,80,81,82]. Similar observations on the lack of activity of checkpoint inhibitors come from studies in germ-free animals that resemble the effect of gut microbiome eradication with the use of antibiotics. Transplantation of fecal microbiota from immune-responding melanoma patients into germ-free mice leads to improved tumor control, augmented T cell responses, and greater efficacy of anti-PD-L1 therapy [83]. Similarly, fecal transplantation from responding melanoma patients into germ-free or antibiotic-pretreated mice could restore antitumor immune responses and induce rejection of implanted tumors, which was not observed when such animals were transplanted with fecal microbiota from non-responding patients [84]. The observation in animal models has been recently replicated in humans. In a phase II clinical trial, 15 melanoma patients refractory to anti PD-1 therapy received responder-derived fecal microbiota transplantation combined with pembrolizumab. Of the 15 treated patients, 6 achieved clinical benefit (one partial response and five disease stabilization) and durable changes in microbiome composition. Responding patients demonstrated an increased abundance of taxa known to be associated with response to anti-PD-1, increased CD8+ T cell activation, and decreased frequency of interleukin-8-expressing myeloid cells [85].

Escherichia coli, a common species found in benign prostate hyperplasia (BPH) tissue, induces inflammation and tissue damage leading to neoplastic transformation in prostate epithelial cells. Ma et al. demonstrated that specific microbes, such *as Listeria monocytogenes, Lactobacillus crispatus,* and *Thermus thermophilus*, prevented tumor formation and growth, while other microbes, such as *Nevskia ramosa* and *S. aureus*, displayed cancer-promoting properties by inducing inflammation, immunosuppression, and promoting prostate cancer stem cells (PCSC) responsible for the development of metastases [86]. Ma et al. have demonstrated that particular bacterial species—Listeria monocytogenes, Methylobacterium radiotolerans *JCM 2831*, *Xanthomonas albilineans GPE PC73*, and *Bradyrhizobium japonicum*—correlated respectively with well-differentiated phenotype, earlier disease stage, lower PSA level, and lower AR expression. A particular species of *Bacteroides*, which is a genus of Gram-negative, anaerobic bacteria, may contribute to the development of high-risk prostate cancer.

The impact of metronomic chemotherapy and its difference from MTD-based microchemotherapy on the gut and intratumoral microbiome is not well recognized yet. An observational study by Zhu et al. suggested that gut microbiota diversity is higher in breast cancer patients than in healthy controls [87], but additional studies demonstrated that the diversity could be decreased with the use of metronomic capecitabine, but not MTD-dosed chemotherapy [88]. Since the impact of MC on tumors and host microbiomes can represent a novel mechanism action of MC, we have initiated a prospective analysis of the impact of various metronomic chemotherapy regimens in breast, ovarian, and prostate cancer patients, and the preliminary results shall be posted soon.

## 4. Metronomic Chemotherapy in Prostate Cancer

Various metronomic chemotherapy-based regimens have been evaluated in advanced PC patients over the last three decades. However, most studies were conducted before the introduction of novel hormonal or cytotoxic agents such as abiraterone acetate, enzalutamide, apalutamide, darolutamide, or cabazitaxel. All the above agents have been approved for the treatment of CRPC patients based on large, well-conducted phase III clinical trials. In comparison, the benefit of MC has been usually evaluated in small, non-randomized trials or retrospective analyses. Accordingly, metronomic chemotherapy should not be considered as a standard option at the early stages of mCRPC, if highly active, well-tolerated novel hormonal agents are freely available. Therefore, MC represents a therapeutic option in mCRPC who failed all available therapies, are unfit for standard chemotherapies, or have no access to standard mCRPC treatments, e.g., in low- or middle-income countries [89]. Nevertheless, many studies demonstrated the activity of MC in advanced mCRPC patients before the advent of novel, active therapies dedicated for this population. Selected studies are summarized in Table 1. In addition to showing the antitumor activity of MC, which was reflected by biochemical responses (PSA decline) in 30–50% of patients, all of the summarized studies demonstrated a very favorable safety profile of MC. This observation underscores the clinical utility of MC, especially in elderly and frail patients, who represent the majority of mCRPC patients. In a recent study, Calvani et al. demonstrated that metronomic cyclophosphamide (50 mg/d p.o.) combined with corticosteroids (dexamethasone 1 mg/d p.o. or prednisone 10 mg/d p.o.) induced biochemical responses (at least 50% PSA reduction) in half of the studied population. However, only two analyzed patients could be considered standard contemporary candidates (pretreatment with docetaxel and novel hormonal agents) for metronomic chemotherapy. Therefore, it is still challenging to draw any conclusions regarding the actual utility of MC in heavily pretreated mCRPC patients since many of the old studies included mCRPC patients for whom, nowadays, at least 1–2 standard treatment options are routinely available. Additionally, most of the listed studies combined MC with steroids (especially dexamethasone), which further complicates the validation of the plain MC activity in mCRPC patients. Several studies have recently demonstrated that a simple switch from prednisone to dexamethasone in mCRPC patients treated with an abiraterone + prednisone combination can lead to profound biochemical responses [90,91,92]. This observation underscores the risk of overestimating the benefit of MC, especially when patients are simultaneously receiving dexamethasone as a part of an antitumor treatment strategy for the first time. There is only a single study evaluating the activity of metronomic chemotherapy in pretreated (docetaxel and ≥1 novel endocrine agent) mCRPC patients. In such mCRPC patients, single-agent cyclophosphamide led to a relatively low rate of PSA responses (16%), with PSF and OS of 4 and 8.1 months, respectively. This study demonstrates that the promising activity of MC observed before the era of novel therapies may not be easily reproducible in contemporary, standardly pretreated mCRPC patients.

## 5. Our Hunt for an Optimal MC Regimen in mCRPC Patients

We have conducted a prospective analysis of consecutive cases of patients treated with various MC regimens to find the most optimal MC combination for a subsequent phase II study. In order to define the real benefit of MC, none of the mCRPC patients was allowed to receive corticosteroids but all continued on pharmacological castration with LHRH analogs. The analysis included nine heavily pretreated patients (median two lines of systemic treatment for mCRPC) and the majority of patients were initially diagnosed with aggressive PC (median Gleason score of 9). Patients received the following combinations—(i) cyclophosphamide 50 mg/d p.o. + vinorelbine 30 mg q2d p.o. (two patients), (ii) cisplatin 25 mg/m^2^ i.v. qw + cyclophosphamide 50 mg/d p.o. (two patients), (iii) paclitaxel 60 mg/m^2^ qw + capecitabine 1500–2000 mg/d (depending on patient weight) p.o. (two patients), (iv) paclitaxel 60 mg/m^2^ qw + capecitabine 1500 mg/d p.o + cyclophosphamide 50 mg/d (three patients). The biochemical responses (PSA decline by ≥50%) were observed in 0%, 50%, 100%, and 100% of patients, respectively. The patients tolerated all analyzed regimens very well, with G1–2 myelotoxicity being the most common adverse event. Based on the prospective analysis of consecutive cases of mCRPC patients treated with MC, the paclitaxel + capecitabine + cyclophosphamide combination has been chosen for further evaluation in a planned phase II clinical trial.

The choice of paclitaxel as a compound for our MC regimen requires explanation. Although cabazitaxel (the new-generation taxoid) is an active and approved agent for the treatment of docetaxel-pretreated mCRPC [50,107,108] patients and studies on standard-dose of paclitaxel failed to demonstrate its activity in prostate cancer [109]; some recent intriguing data justify the choice of paclitaxel as an element of our MC regimen. It is well known that paclitaxel, administered weekly, exerts robust antiangiogenic activity [110,111,112]. Our choice of paclitaxel in mCRPC is also justified by a recent study comparing weekly paclitaxel with cabazitaxel in advanced breast cancer. The study aimed to demonstrate the superiority of cabazitaxel 25 mg/m^2^ q3w over weekly paclitaxel as a first-line treatment in 158 patients with metastatic breast cancer, many of whom have previously received docetaxel in the neoadjuvant or adjuvant setting. Surprisingly, both drugs demonstrated similar OS, PFS, and ORR [113], thus suggesting the need for reevaluation of weekly paclitaxel in docetaxel-pretreated mCRPC patients.

## 6. Conclusions

Metronomic chemotherapy represents a unique treatment option for many advanced cancer patients, including those diagnosed with mCRPC. However, it should be not considered as a first-line option in chemotherapy-fit mCRPC patients due to the lack of robust, prospective data suggesting its equality to standard treatment options, including novel antiandrogen agents (abiraterone, enzalutamide, darolutamide, and apalutamide) [4,6,7,8,114], chemotherapeutic agents (docetaxel and cabazitaxel) [21,22,115], or novel targeted therapies (olaparib in HRD mCRPC or Lutetium-177–PSMA-617) [18,116]. The use of MC based on the current state of knowledge should be restricted, in a typical population of mCRPC patients, to later lines of treatment, where no standard treatment options nor dedicated clinical trials are available. Moreover, recent disappointing data on the lack of activity of novel immunotherapies in mCRPC when administered as single agents [117] represent an intriguing hint to combining checkpoint inhibitors with immunomodulating MC regimens. However, the MC remains an option for earlier lines of therapy in fragile, elderly patients in whom standard treatment options (especially bi- or three-weekly chemotherapy regimens) may be intolerable. There is also no doubt that MC, with its multidirectional mechanism of action, represents a treatment option for mCRPC patients in low- and middle-income patients who do not have access to still relatively novel and expensive anticancer agents. Therefore, studies on metronomic chemotherapy in various clinical settings are critical from a global perspective.

## Figures and Tables

**Table 1 jcm-11-02853-t001:** Studies evaluating metronomic chemotherapy in prostate cancer.

Regmien	Number of Patients	Biochemical Response (>50% PSA Reduction)	PFS/OS (Months)	Ref
CTX 50 mg p.o. + DEX 1 mg p.o.	34	39%	NR/NR	[93]
CTX 50 mg p.o.	58	34.5% ^1^	NR/NR	[94]
CTX 500 mg/m^2^ i.v. induction (day 1.) → CTX 50 mg/d p.o. + CXB 200 mg p.o. bid + DEX 1 mg/d p.o.	28	32%	3.0/21.0	[95]
CTX 50 mg p.o. + DEX 1 mg p.o.	17	24%	NR/24.0	[96]
CTX 50–100 mg p.o. + prednisone 10 mg/d p.o.	18	44% ^1^	4.7/NR	[97]
CTX 50 mg p.o. + prednisone 10 mg/d p.o.	23	26%	6.0/11.0	[98]
VRB 25 mg/m^2^ iv 12× qw → q2w + prednisone 10 mg/d p.o.	14	36%	4.5/17	[99]
CTX 50 mg/d + MTX 2.4 mg po twice a week	58	25%	5.2/11.5	[100]
CTX 100 mg/d p.o. UFT 400 mg/d p.o. DEX 1 mg/d p.o.	57	63%	7.2/NR	[101]
CAP 1000 mg bid p.o. d 1–14 q21 + CTX 50 mg/d p.o. + thalidomide 100 mg/d p.o. + prednisone 5 mg bid p.o.	28	35.7%	4.7/19.5	[102]
DXL 35 mg/m^2^ i.v. qw + CAP 625 mg/m^2^ bid d 5–18 q4w (4 cycles)	44	68%	NR/17.7	[103]
CTX 50 mg/d p.o. + DEX 1 mg/d p.o.	24	33%	5.0/19.0	[104,105]
CTX 50 mg/d p.o. + DEX 1 mg/d p.o.	37	51%	11.0/28.0	[105,106]
CTX 50 mg/d p.o. ^2^	74	16%	3.0/7.5	[106]

CTX—cyclophosphamide, VRB—vinorelbine, MTX—methotrexate, UFT—Tegafur/uracil, DXL—docetaxel, CAP—capecitabine, DEX—dexamethasone, PFS—progression-free survival, OS—overall survival, NR—not reported, ^1^—any PSA decrease, p.o.—oral administration, i.v.—intravenous administration, bid-twice daily, ^2^—typical, contemporary mCRPC patients (pretreated with DXL and novel endocrine agents).

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
