# Peer review of "Metronomic Chemotherapy in Prostate Cancer"

_jcm, 2022, doi:10.3390/jcm11102853_

Round 1
Reviewer 1 Report
The concept of MC has been well introduced in this review article. Comparison on the difference between traditional MTD chemotherapy and MC in various aspects like the angiogenesis inhibition, immunological effects, action on cancer stem cells and the microbiome effect has been clearly presented. It would be more interesting to include more details in how the MC impact on the cancer stem cell section.
Since MC is regarded as the therapeutic option in mCRPC which failed other available therapics and standard chemotherapies, would the author comment on any possible ways to evaluate the actual effect of MC in the heavily pretreated mCRPC cases. Furthermore, would the biochemical response in the summary table considered as rather low or it could be presented with the response for tradition treatment result in parallel if possible.
Author Response
Some additional data on the differential impact of MTD and MC on tumor-initiating stem cells has been added (reference 65).
Reviewer 2 Report
The manuscript by Wysocki and co-workers summarizes the available data on metronomic chemotherapy in prostate cancer. The authors briefly describe the most important mechanisms of action of metronomic chemotherapy impacting on tumour cell proliferation, angiogenesis, immunosuppression, cancer stem cell functions, as well as tumor microbiome. Moreover, based on prospective analysis in metastatic castration-resistant prostate cancer patients unresponsive to available standard therapies, they selected the metronomic combination of paclitaxel, capecitabine and cyclophosphamide as a preferred regimen for a phase II study for these men.The review is well written, easy to follow and unaffected by important flaws.
Minor points
Please correct typos.
Paragraph 6 should be Conclusions
Author Response
The final section has been modified according to reviewer's comment - summary paragraph was renamed as conclusions